# Evaluation of a Dietary Supplementation Combining Protein and a Pomegranate Extract in Older People: A Safety Study

**DOI:** 10.3390/nu14235182

**Published:** 2022-12-06

**Authors:** Valérie Dormal, Barbara Pachikian, Elena Debock, Marine Buchet, Sylvie Copine, Louise Deldicque

**Affiliations:** 1Center of Investigation in Clinical Nutrition, Université Catholique de Louvain, 1348 Louvain-la-Neuve, Belgium; 2Institute of Neuroscience, Université Catholique de Louvain, 1348 Louvain-la-Neuve, Belgium

**Keywords:** protein, pomegranate extract, older people, safety, clinical trial

## Abstract

Malnutrition is a highly prevalent condition in older adults. It is associated with low muscle mass and function and increased occurrence of health problems. Maintaining an adequate nutritional status as well as a sufficient nutrient intake in older people is therefore essential to address this public health problem. For this purpose, protein supplementation is known to prevent the loss of muscle mass during aging, and the consumption of various pomegranate extracts induces numerous health benefits, mainly through their antioxidant properties. However, to our knowledge, no study has to date investigated the impact of their combination on the level of malnutrition in older people. The objective of this preliminary study was thus to evaluate the safety of a combination of protein and a pomegranate extract in healthy subjects aged 65 years or more during a 21-day supplementation period. Thirty older participants were randomly assigned to receive protein and a pomegranate extract (Test group) or protein and maltodextrin (Control group) during a 21-day intervention period. The primary outcomes were the safety and tolerability of the supplementation defined as the occurrence of adverse events, and additional secondary outcomes included physical examination and hematological and biochemical parameters. No serious adverse events were reported in any group. Changes in physical, hematological, and biochemical parameters between the initial screening and the end of the study were equivalent in both groups, except for glutamate-pyruvate transaminase (GPT) and prealbumin, for which a decrease was observed only in the Test group. Our initial findings support the safety of the combination of protein and a pomegranate extract in healthy elderly people. Future clinical trials on a larger sample and a longer period are needed to determine the efficacy of this combination.

## 1. Introduction

Malnutrition in older adults has been recently identified as a challenging public health problem associated not only with a high mortality and morbidity rate but also with physical, psychological, and functional decline [1,2]. In particular, protein–energy malnutrition (PEM), resulting from a decrease in protein and energy intake, has been associated with a number of health problems such as sarcopenia [3,4]. Sarcopenia is characterized by a gradual loss of muscle mass leading to a decrease in muscle strength and physical performance. This condition can have functional consequences, such as falls and fractures, and worsens the health and the quality of life of the elderly [5].

The treatment of malnutrition requires early identification and efficient intervention, both in hospitalized patients and community-dwelling older adults. Higher protein intake recommendations are made to older adults suffering from malnutrition. Beyond its role in the accretion of muscle mass, protein has been identified as a key nutrient for elderly people. Slightly higher protein intake than usually recommended may improve muscle health and help maintain energy balance, weight management, and cardiovascular function [6], all of which may contribute to an improved quality of life. Compared to the typical recommended protein intake of 0.83 g·kg^−1^·d^−1^ in healthy adults, the recommended intake rises to 1 g·kg^−1^·d^−1^ in older malnourished persons with a body mass index (BMI) under 21 kg/m^2^ [7]. Currently, the most common method to reach that recommendation is consuming oral nutritional supplements (ONS). Several clinical studies in elderly people have shown that protein- and vitamin-D-rich ONS improve protein synthesis, muscle mass, and performance but also functional parameters, evaluated, for example, by the chair rise test [8,9,10]. However, evidence investigating ONS compliance and effectiveness is conflicting [11]. In addition, those supplements are rarely effectively consumed by the elderly because of their taste, their satietogenic texture, and/or their high cost [12].

More tolerable and appropriate alternatives that meet the needs of malnourished older people are therefore required. Besides integrating protein directly into the food consumed, the combination with other substances beneficial to the elderly may constitute another strategy. In this study, protein of high biological value was combined with a pomegranate extract, a rich source of various phytochemicals, including anthocyanins, ellagitannins, gallotannins, proanthocyanidins, flavonols, and lignans, which are known to induce beneficial effects on human health [13]. The pomegranate extract used here was particularly rich in ellagitannins, more specifically in punicalagins, as well as in ellagic acid. Thanks to its antioxidant and anti-inflammatory mechanisms, pomegranate is known to improve muscle function [14] and performance [15] in resistance-trained men. In addition to the positive effects of pomegranate on muscles, pomegranate also has other very interesting properties, including protection against metabolic and cardiovascular diseases [16,17]. Pomegranate polyphenols are able to inhibit low-density lipoprotein (LDL) oxidation and to increase the activity of serum paraoxonase, an esterase that protects lipids against peroxidation in humans [18]. Here, we chose to test whey protein, a protein source of high biological value due to its high content of essential amino acids, particularly leucine. This amino acid is particularly effective at stimulating protein synthesis in the skeletal muscle [19]. Whey protein has proven to be a very efficient protein source to stimulate protein synthesis and the accretion of muscle mass in diverse populations, including older people [20,21]. 

Altogether, these data show that protein and pomegranate extract supplementation are effective nutritional strategies to improve health status, at least when taken separately. Whether there is an added value in combining them and whether pomegranate extracts provide health benefits in older persons is unknown. Combining protein and a pomegranate extract would be innovative but safety issues may arise as their combination has never been tested in an older population. The aim of this study was therefore to evaluate the safety of a 21-day supplementation combining protein and a pomegranate extract in healthy subjects aged 65 years or more compared to protein alone.

## 2. Materials and Methods

### 2.1. Participants

A total of 39 participants were assessed for eligibility, and 30 participants were randomized between December 2021 and February 2022 into the Test (*n* = 15) or the Control (*n* = 15) groups (Figure 1). One participant from the Test group dropped out of the study for personal reasons. Participants were recruited by posters, mail, social networks, and local newspapers. To be included, the participants had to meet the following criteria: woman or man aged over 65 years; presenting a BMI from 20 to 30; and in good general health as evidenced by medical history and physical examination. The participants were excluded if they presented one of the following exclusion criteria: severe psychiatric disorder or disease within 6 months before inclusion (such as depression, bipolar disorder, severe anxiety, psychosis, schizophrenic disorders, or dementia) or a severe neurologic trouble (such as Alzheimer’s, autism, or Parkinson’s disease); cancer within less than 3 years before the screening visit (except basal cell skin cancer or squamous cell skin cancer); a severe gastro-intestinal, hepatic, respiratory, kidney, or cardiovascular disorder or severe infection (such as HIV or hepatitis); uncontrolled hormonal disorders (such as thyroid problems or Cushing’s syndrome); uncontrolled type 1 or 2 diabetes; known allergy or intolerance to one of the components of the administered products; and participants consuming ONS or protein supplements up to one month before the screening. 

All selected participants provided written informed consent. The trial was approved by the local ethical committee and was carried out in accordance with the Declaration of Helsinki and the Good Clinical Practice guidelines as required by the following regulations: the Belgian law of 7 May 2004 regarding experiments on human beings and the EU Directive 2001/20/EC on Clinical Trials (registration at clinicaltrials.gov accessed on 6 November 2022 as NCT05527249).

### 2.2. Study Design and Randomization

This was a randomized double-blind placebo-controlled interventional study. A prescreening was proposed with an online questionnaire sent by email. Then, the screening visit, comprising a physical examination (including measurement of body weight, height, heart rate, and blood pressure) and a blood sampling, was organized within 4 weeks before subject inclusion. The participants who met all the criteria were randomly assigned to the Test or the Control groups. Both groups were instructed (1) to consume 20 g protein mixed into vegetable milk, orange juice, or soup and (2) to ingest one capsule with a glass of water every day before lunch for 3 weeks. The participants of both groups received the same protein powder (Fresubin^®^) with a neutral taste and containing 100% whey protein. The capsule contained 650 mg pomegranate extract (Oxylent GR) in the Test group and 650 mg maltodextrin (Delical, BS Nutrition) in the Control group. Each capsule of pomegranate extract contained at least 65 mg punicalagins (>10%), 260 mg polyphenols (>40%), and more than 13 mg ellagic acid (>2%). The dose of 650 mg was chosen based on the efficacy of this dose to improve muscle strength recovery after eccentric exercise [22]. The capsules containing maltodextrin or pomegranate extract were prepared by a pharmacist. The appearance, i.e., white color, did not distinguish the 2 capsules. All products were stored in a closed room at a controlled temperature between 18 and 25 °C and protected from light. Study outcomes were assessed at baseline (V1), in the middle of the intervention (day 10; V2), and at the end of the intervention (day 21; V3). Protocol deviations and subjects’ withdrawals were monitored regularly during the study. The database was locked, and the blinding was broken after completion of the whole quality control of the data.

### 2.3. Safety Measurements

The primary outcomes were safety and tolerability of the protein/pomegranate combination defined as occurrence of adverse events. All these events were recorded from the signature of the informed consent to the end of the study. 

The secondary outcomes were changes in physical examination, including weight, heart rate, and systolic and diastolic blood pressure; changes in hematological parameters, including hemoglobin and hematocrit levels and white blood cell, red blood cell, and platelet counts; and changes in biochemical parameters (Lims Mbnext group), including uric acid, creatinine, total bilirubin, glutamate oxaloacetic transaminase (GOT), glutamate pyruvate transaminase (GPT), total cholesterol, triglyceride, sodium, potassium, total protein, prealbumin, urea, blood glucose, alkaline phosphatase, gamma-glutamytranspeptidase (GGT), lactate dehydrogenase (LDH), and C-reactive protein (CRP). Compliance to protein and pomegranate/maltodextrin ingestion was assessed at the end of the study by weighing the remaining quantity of powder and capsules and comparing to the quantity given at the beginning of the study. 

### 2.4. Statistical Analyses

Given the pilot nature of this study, a sample size of 30 participants was determined based on feasibility considerations. All the continuous variables are presented as mean ± SD. Statistical analyses were performed using the software systems SAS 9.4 (SAS Institute Inc., Cary, NC, USA). Baseline differences in demographic characteristics between the groups were examined using χ² tests and independent t tests. Differences in physical, hematological, and biochemical parameters between groups (Control and Test) at each visit (V1, V2, and V3) were analyzed using a linear mixed model for repeated measurements with subjects as the random variable, and groups, visits, and their interaction as fixed independent variables. As variables were not normally distributed, a logarithmic transformation was applied, and when appropriate, post hoc comparison was performed using the Wilcoxon matched-pairs signed-rank test for analyses of within-group differences and the Wilcoxon Mann–Whitney two-sample test for analyses of differences between groups. All analyses were conducted at an alpha level of 0.05.

## 3. Results

### 3.1. Participants’ Characteristics

The Test group was composed of 5 men and 9 women, and the mean age was 68.4 ± 2.4 years, while the Control group was composed of 9 men and 6 women, and the mean age was 68.7 ± 2.8 years. There were no significant differences in age, gender ratio, weight, BMI, or blood pressure between the two groups at baseline.

### 3.2. Primary Outcomes

The protein/pomegranate combination was well tolerated, with no serious adverse event reported and no difference in the mild and moderate adverse events reported by the Test (*n* = 5) and the Control (*n* = 6) groups. None of these mild or moderate adverse effects, including lung infection, COVID-19, and painful tooth extraction, were deemed to be related to the supplementation (Table 1).

### 3.3. Secondary Outcomes

Data related to physical, hematological, and biochemical parameters are presented in Table 2 and Figure 2 and Figure 3.

#### 3.3.1. Physical Parameters

The systolic (−4.2%, *p* = 0.046, Figure 2A) and diastolic (−7.5%, *p* = 0.003, Figure 2B) blood pressures decreased between V1 and V3 in both the Control and Test groups, with no difference between the two groups. No difference between conditions was observed in the weight, the BMI, or the heart rate (Table 2). 

#### 3.3.2. Hematological Parameters

Plasma hemoglobin levels decreased between V1 and V3 in both groups (−1.8%, *p* = 0.038), with no difference between groups (Table 2). No difference between conditions was observed in the hematocrit levels or the white blood cell, red blood cell, or platelet counts. 

#### 3.3.3. Biochemical Parameters

Plasma GPT levels decreased between V1 and V3 only in the Test group (−15.7%, *p* = 0.039, Figure 3A). Total plasma cholesterol levels decreased between V1 and V3 in both groups (−6.9%, *p* < 0.001), with no difference between groups (Figure 3B). Plasma prealbumin levels decreased between V1 and V3 in the Test group (−12.1%, *p* < 0.001) and tended to decrease in the Control group (−3.6%, *p* = 0.071, Figure 3C). In both groups, the levels of plasma uric acid decreased between V1 and V2 (−9.3%, *p* < 0.001) and V1 and V3 (−10.6%, *p* < 0.0001) while the levels of urea increased at the same time between V1 and V2 (+12.5%, *p* = 0.011), with no difference between groups (Table 1). Plasma potassium levels increased similarly in both groups between V1 and V2 (+8.9%, *p* < 0.001) and between V1 and V3 (+4.9%, *p* < 0.001). Finally, plasma creatinine, total bilirubin, GOT, triglyceride, sodium, protein, blood glucose, alkaline phosphatase, GGT, LDH, and CRP levels were stable throughout the study and did not differ among groups (Table 2).

#### 3.3.4. Compliance

A high compliance percentage in both groups was found, with an intake of 100.0 ± 5.7% in the Test group and 100.0 ± 6.8% in the Control group, with no difference between the two groups (Appendix A).

## 4. Discussion

Malnutrition is a highly prevalent condition in older adults. It is associated with low muscle mass and function and increased occurrence of health problems [1,2,3,4]. Maintaining an adequate nutritional status as well as a sufficient nutrient intake in older people is therefore essential to address this public health problem. For this purpose, protein supplementation is known to prevent the loss of muscle mass during aging [8,9,10], and the consumption of various pomegranate extracts induces numerous health benefits, mainly through their antioxidant properties [13]. Combining protein and a pomegranate extract would be innovative but safety issues may arise as this combination has never been tested in an older population. The aim of this study was therefore to evaluate the safety of a 21-day supplementation combining protein and a pomegranate extract in healthy subjects aged 65 years or more compared to protein alone.

Our results showed that the daily combination of 20 g protein and 650 mg pomegranate extract for 21 days was safe and well tolerated. No serious adverse events were observed, and all mild or moderate adverse events were not related to the dietary supplement consumption. These results reinforce previous studies showing the safety of pomegranate extract supplementation at similar or higher doses in different adult populations, e.g., 1050 mg in hemodialysis patients [23] or 710 and 1420 mg in overweight individuals [24]. In addition, there were no differences between the Test and Control groups in physical and hematological parameters. In line with previous results dealing with protein or pomegranate supplementation [25,26], a decrease in systolic and diastolic blood pressure was observed between the start and the end of the study in both groups. While beyond the primary scope of the present study, these results suggest beneficial actions of protein and protein/pomegranate supplementation on blood pressure. To confirm these preliminary observations, a proper control group with neither protein nor pomegranate supplementation needs to be included in our follow-up study investigating the efficacy of those compounds on physical and hematological parameters in malnourished older people.

Most of the differences in the biochemical parameters observed between baseline screening and the end of the study were present in both groups. Plasma potassium and urea levels increased while uric acid and total cholesterol decreased similarly in both groups, which is consistent with previous reports following protein intake in healthy and pathological populations [27,28,29]. Here, we did not find any additional effect of combined protein/pomegranate supplementation compared to protein alone on the different biochemical parameters studied, possibly excepting GPT, a marker of liver function [30]. The latter decreased in the Test group only. In addition to its already known effects on metabolic risk factors [16], pomegranate could have a potential beneficial effect on parameters related to liver function. Here as well, a proper control group with neither protein nor pomegranate supplementation will be necessary when investigating the efficacy of those compounds on biochemical parameters in malnourished older people. Globally, given the preliminary nature of this safety study, the small number of participants resulted in limited statistical power to demonstrate specific benefits of protein and pomegranate on health-related parameters. That said, the primary aim of the present study was fulfilled, namely, the investigation of the safety and tolerance of combined protein/pomegranate supplementation in a limited sample of older people. Despite some statistical changes, all parameters studied here remained within healthy thresholds. Therefore, our supplementation can be considered safe and well tolerated.

In conclusion, we found that a combined protein/pomegranate supplementation for 21 days was safe and well tolerated in older participants. This first study was required before investigating the efficacy of this combination in a larger sample of older malnourished people for a longer duration to detect changes in muscle mass, and in a more appropriate form such as directly integrated into food to increase ONS compliance.

## Figures and Tables

**Figure 1 nutrients-14-05182-f001:**
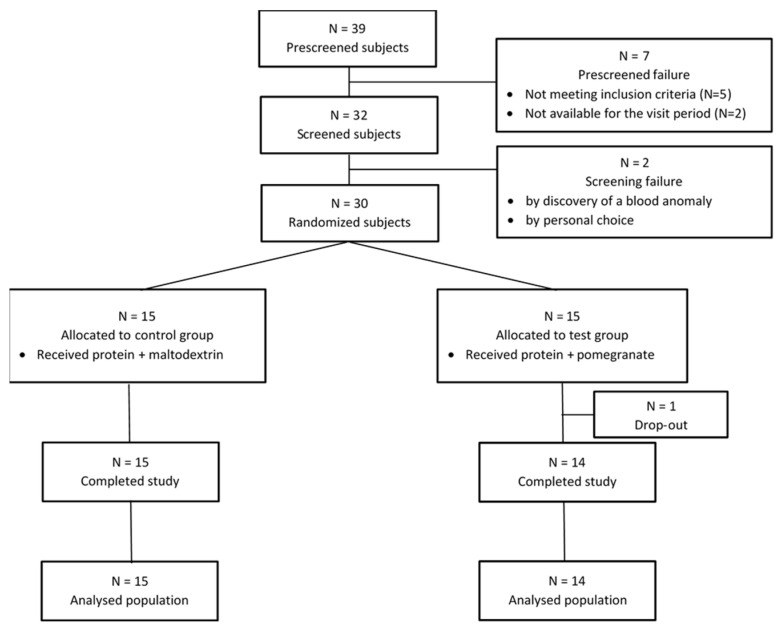
Flow chart of the study samples, including the number of participants who were screened, underwent randomization, completed the study treatment, and were analyzed for the primary and secondary outcomes.

**Figure 2 nutrients-14-05182-f002:**
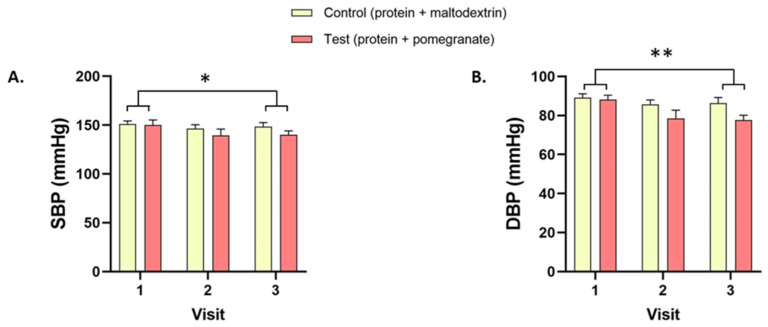
Systolic and diastolic blood pressures. (**A**) Systolic (SBP) and (**B**) diastolic (DBP) blood pressures in the Test and Control groups at V1, V2, and V3. * *p* < 0.05, ** *p* < 0.01.

**Figure 3 nutrients-14-05182-f003:**
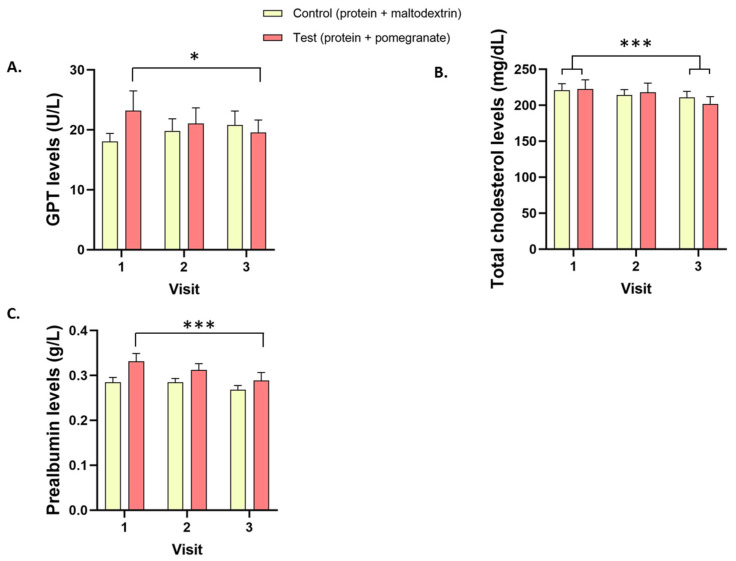
Glutamate pyruvate transaminase, total cholesterol, and prealbumin levels. Plasma (**A**) glutamate pyruvate transaminase (GPT), (**B**) total cholesterol, and (**C**) prealbumin levels in the Test and Control groups at V1, V2, and V3. * *p* < 0.05, *** *p* < 0.001.

**Table 1 nutrients-14-05182-t001:** Description of mild and moderate adverse events observed in the Test and Control groups.

	Description of the Adverse Event	Intensity of the Event	Causal Relation of the Event to the Study Product
**Test**	COVID	Moderate	Not related
	Intestinal problems	Moderate	Maybe related
	Intestinal problems	Moderate	Unlikely related
	Headache	Mild	Not related
	Tendinitis	Moderate	Not related
**Control**	Lung infection	Moderate	Not related
	COVID	Mild	Not related
	COVID	Moderate	Not related
	Eczema	Moderate	Unlikely related
	Sciatic nerve inflammation	Moderate	Not related
	Painful tooth extraction	Moderate	Not related

**Table 2 nutrients-14-05182-t002:** Physical, hematological, and biochemical parameters.

	Test (*n* = 14)			Control (*n* = 15)		
	V1	V2	V3	V1	V2	V3
**Physical parameters**						
Weight (kg)	70.9 ± 14.7	72.8 ± 16.7	72.1 ± 16.0	72.3 ± 10.6	70.2 ± 10.4	72.3 ± 11.6
Heart rate (bpm)	64.9 ± 6.9	66.6 ± 11.4	71.1 ± 12.0	65.3 ± 11.0	67.4 ± 17.7	65.9 ± 12.3
Body Mass Index (kg/m^2^)	25.7 ± 3.2	26.8 ± 3.5	26.5 ± 3.3	24.7 ± 2.2	24.3 ± 2.5	25.0 ± 3.1
**Hematological parameters**						
White blood cells (10^3^/μL)	5.89 ± 1.38	5.82 ± 1.24	5.83 ± 1.13	5.57 ± 1.01	5.83 ± 1.37	5.50 ± 1.21
Red blood cells (10^6^/μL)	4.62 ± 0.30	4.59 ± 0.29	4.53 ± 0.26	4.68 ± 0.42	4.69 ± 0.35	4.64 ± 0.35
Hemoglobin (g/dL)	13.9 ± 1.2	13.7 ± 0.9	13.6 ± 0.9 *	14.1 ± 1.1	14.0 ± 1.0	14.0 ± 1.0 *
Hematocrit (%)	41.0 ± 2.8	40.6 ± 2.4	40.2 ± 2.4	41.3 ± 3.1	41.4 ± 2.8	41.3 ± 2.7
Platelet count (10^3^/μL)	267.6 ± 40.6	265.1 ± 47.7	265.6 ± 36.8	241.8 ± 46.6	248.3 ± 45.3	245.7 ± 42.9
**Biochemical parameters**						
Uric acid (mg/dL)	5.20 ± 1.26	4.76 ± 1.12 ***	4.59 ± 1.07 ***	5.52 ± 1.33	4.96 ± 1.39 ***	4.99 ± 1.17 ***
Creatinine (mg/dL)	0.80 ± 0.17	0.77 ± 0.12	0.74 ± 0.15	0.81 ± 0.15	0.80 ± 0.16	0.80 ± 0.17
Total bilirubin (mg/dL)	0.77 ± 0.27	0.72 ± 0.24	0.69 ± 0.26	0.84 ± 0.40	0.79 ± 0.39	0.82 ± 0.41
GOT (U/L)	23.4 ± 6.6	22.6 ± 5.5	21.4 ± 4.0	22.6 ± 4.2	23.6 ± 9.7	23.7 ± 5.8
Triglycerides (mg/dL)	121.9 ± 72.5	109.2 ± 45.9	106.8 ± 59.9	98.2 ± 43.6	99.1 ± 48.9	94.5 ± 44.2
Sodium (mmol/L)	139.6 ± 2.3	139.7 ± 2.5	140.1 ± 2.1	139.5 ± 2.0	139.5 ± 2.8	139.5 ± 2.2
Potassium (mmol/L)	4.32 ± 0.30	4.68 ± 0.37 ***	4.59 ± 0.37 ***	4.37 ± 0.21	4.78 ± 0.36 ***	4.53 ± 0.40 ***
Total protein (g/L)	68.4 ± 3.7	68.0 ± 3.1	66.5 ± 3.7	66.9 ± 3.0	67.2 ± 2.7	66.8 ± 3.3
Urea (mg/dL)	37.1 ± 10.4	42.1 ± 10.0 *	40.1 ± 6.1	35.7 ± 10.2	39.7 ± 9.4 *	37.7 ± 7.8
Blood glucose (mg/dL)	89.3 ± 9.2	87.6 ± 8.1	90.1 ± 11.5	89.1 ± 8.6	88.1 ± 7.3	89.9 ± 9.6
Alkaline phosphatase (U/L)	69.1 ± 22.9	68.4 ± 21.4	68.5 ± 20.8	68.8 ± 15.4	68.0 ± 13.8	68.3 ± 12.6
GGT (U/L)	27.4 ± 19.2	25.2 ± 16.8	24.6 ± 15.2	23.3 ± 8.5	24.2 ± 11.7	24.7 ± 14.4
LDH (U/L)	168.1 ± 21.1	168.9 ± 21.8	166.5 ± 26.6	175.2 ± 29.6	169.5 ± 24.2	174.2 ± 23.3
CRP (mg/L)	2.26 ± 1.75	2.71 ± 1.43	3.88 ± 6.86	1.79 ± 1.12	1.55 ± 0.85	1.65 ± 1.21

Values are means ± SD. n: number of subjects; bpm: beats per minute; GOT: glutamate oxaloacetic transaminase; GGT: gamma-glutamyltranspeptidase; LDH: lactate dehydrogenase; CRP: C-reactive protein. * *p* < 0.05 vs. V1, *** *p* < 0.001 vs. V1.

## Data Availability

Data are available upon request by sending an e-mail to cicn@uclouvain.be.

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
