# Peer review of "Evaluation of a Dietary Supplementation Combining Protein and a Pomegranate Extract in Older People: A Safety Study"

_nutrients, 2022, doi:10.3390/nu14235182_

Round 1

Reviewer 1 Report

Intro:

In the introduction, the authors introduced the function of protein which helps the reader to get the importance of it. However, in the current study, it would be more relevant to introduce the specific protein used in the trial and specify why it was chosen (there are many other more, why this one?).

In addition, the authors nicely pointed that the pomegranate extra contains many compounds of beneficial effects on human. Any information already known about its (the pomegranate extract) impact on human already? In other words, any intervention has been done to test its effect on human? Is it the same extract as in reference 15?

M&M

How is the dosage, in terms of pomegranate extract, decided?

Results

Line 149: “ There were no difference…..” should be “There were no significant difference…..”

Line 198-199, would be nice to include the compliance data in supplementary.

Discussion

Nice discussion regarding to the decrease of blood pressure in line 224-227

Author Response

REVIEWER 1

We thank the reviewer for his/her valuable comments that contributed to improve the quality of our manuscript. Please find our point-by-point response to your comments below.

Intro:

In the introduction, the authors introduced the function of protein which helps the reader to get the importance of it. However, in the current study, it would be more relevant to introduce the specific protein used in the trial and specify why it was chosen (there are many other more, why this one?).

We have now added the following paragraph to the introduction: "Here, we chose to test whey protein, a protein source of high biological value due to its high content in essential amino acids and more particularly in leucine. The latter amino acid is particularly potent to stimulate protein synthesis in the skeletal muscle [18]. Whey protein has proven to be a very efficient protein source to stimulate protein synthesis and the accretion of muscle mass in diverse populations, including older people [19,20]."

In addition, the authors nicely pointed that the pomegranate extra contains many compounds of beneficial effects on human. Any information already known about its (the pomegranate extract) impact on human already? In other words, any intervention has been done to test its effect on human? Is it the same extract as in reference 15?

Many interventional studies have already been carried out in human, which are presented and summarized in diverse systematic reviews such as in Laurindo et al, Nutrients, 2022 specifically looking at the metabolic syndrome, in Asgary et al, Curr Problems Card, 2022 specifically looking at oxidative stress or in Gimenez-Bastida et al, Trends Food Sci Technol, 2021 looking more broadly at health properties of pomegranate juices and extracts.

To answer the second question, it is not the same extract as in reference 15. In Torregrosa-Garcia et al, participants ingested for 15 days two capsules (2x375mg) of pomegranate extract, corresponding in total to at least 225mg punicalagins (>30%) and 375mg polyphenols (>50%) and less than 60mg ellagic acid (<8%) per day. Here, participants ingested for 21 days a pomegranate extract containing at least 65mg punicalagins (>10%), 260mg polyphenols (>40%) and more than 13mg ellagic acid (>2%). We have now added this information in the M&M section: "Each capsule of pomegranate extract contained at least 65mg punicalagins (>10%), 260mg polyphenols (>40%) and more than 13mg ellagic acid (>2%)."

M&M

How is the dosage, in terms of pomegranate extract, decided?

The dosage was chosen based on the literature showing beneficial health effects and interesting effects on muscle performance with a range of dosages of a few hundred milligrams. More particularly, the dose of 650mg was based on the results obtained by Machin et al (Physiol J, 2014) reporting an improved recovery of muscle strength after eccentric exercise compared to a placebo, with no difference between 650mg and 1300mg of gallic acid equivalent. We therefore chose for the lowest efficient dose. We have now added this information to the M&M section: "The dose of 650mg was chosen based on the efficacy of this dose to improve muscle strength recovery after eccentric exercise [21]."

Results

Line 149: “ There were no difference…..” should be “There were no significant difference…..”

The term significant has now been added.

Line 198-199, would be nice to include the compliance data in supplementary.

The individual compliance data have now been added in a supplementary table.

Discussion

Nice discussion regarding to the decrease of blood pressure in line 224-227.
Thank you for those kind words.

Reviewer 2 Report

The manuscript entitled "Evaluation of a dietary supplementation combining protein and a pomegranate extract in older people: a safety study" addresses a current topic. Stores and pharmacies are full of food supplements, but many without results.

The manuscript is well written and organized, but I think more clarifications should be made:

The abstract lacks the clear purpose of this study.

What compounds in pomegranates are tracked and what benefits does it have? What type of proteins, their source

The introduction provides relevant information from the literature, but not enough.

I recommend adding information about the role of proteins, amino acids in the human body / of the elderly.

I recommend the detailing of the information regarding the target compounds in pomegranate.

In the Materials and Methods section, the methods are properly presented, but the material section is missing.

I recommend a section with the material used

Details of the supplement used: raw material used, method of preparation of the nutritional supplement, storage conditions, instructions for use.

The results are presented succinctly, but sufficiently.

In table 2, the statistical analysis is partially carried out. The standard deviation is not sufficient for statistical analysis.

The discussion section needs to be significantly improved.

In this section, the obtained results must be commented and discussed

Lines 201-211 are not relevant to this section

Line 229-232 - "Plasma potassium and urea levels increased while uric acid and total cholesterol decreased similarly in both groups, which is consistent with previous reports following protein intake in healthy and pathological populations" - please explain these results and their consequences. Please explain the results and their consequences for the following results as well.

The aim was to investigate safety and tolerance, but the expected benefits must be clearly established. We will not administer supplements just because they do not harm us. The goal is to do us good.

Author Response

REVIEWER 2

We thank the reviewer for his/her valuable comments that contributed to improve the quality of our manuscript. Please find our point-by-point response to your comments below.

The manuscript entitled "Evaluation of a dietary supplementation combining protein and a pomegranate extract in older people: a safety study" addresses a current topic. Stores and pharmacies are full of food supplements, but many without results.

The manuscript is well written and organized, but I think more clarifications should be made:

The abstract lacks the clear purpose of this study.

The purpose is stated between line 15 and line 17: "The objective of this preliminary study is thus to evaluate the safety of a combination of protein and a pomegranate extract in healthy subjects aged 65 years or more during a 21 day-supplementation period."

What compounds in pomegranates are tracked and what benefits does it have? What type of proteins, their source

We have now added the protein source and the reason for this source: "Here, we chose to test whey protein, a protein source of high biological value due to its high content in essential amino acids and more particularly in leucine. The latter amino acid is particularly potent to stimulate protein synthesis in the skeletal muscle [18]. Whey protein has proven to be a very efficient protein source to stimulate protein synthesis and the accretion of muscle mass in diverse populations, including older people [19,20]."

To answer the first part of the comment, the compounds tracked in pomegranates, i.e. mainly polyphenols, are cited in the introduction as well as their potential health benefits (lines 55-67). In addition, we have now included the specific active compounds found in our pomegranate extract. "In this study, protein of high biological value will be combined with a pomegranate extract, a rich source of various phytochemicals, including anthocyanins, ellagitannins, gallotannins, proanthocyanidins, flavonols, and lignans, which are known to induce beneficial effects on human health [12]. The pomegranate extract used here was particularly rich in ellagitannins, and more specifically in punicalagins, as well as in ellagic acid. Thanks to its antioxidant and anti-inflammatory mechanisms, pomegranate is known to improve muscle function [13] and performance [14] in resistance trained men. In addition to the positive effects of pomegranate on muscles, pomegranate also has other very interesting properties, including protection against metabolic and cardiovascular diseases [15,16]. Pomegranate polyphenols are able to inhibit low density lipoprotein (LDL) oxidation and to increase the activity of serum paraoxonase, an esterase that protects lipids against peroxidation, in human [17]."

The introduction provides relevant information from the literature, but not enough.

I recommend adding information about the role of proteins, amino acids in the human body / of the elderly.

The following information has now been added in the introduction: "Beyond their role in the accretion of muscle mass, protein have been identified as a key nutrient for elderly people. Slightly higher protein intake than usually recommended may improve muscle health, and help maintain energy balance, weight management, and cardiovascular function [6]. All may contribute to an improved quality of life."

I recommend the detailing of the information regarding the target compounds in pomegranate.

The general active compounds in pomegranate are mentioned in the introduction: "In this study, protein of high biological value will be combined with a pomegranate extract, a rich source of various phytochemicals, including anthocyanins, ellagitannins, gallotannins, proanthocyanidins, flavonols, and lignans, which are known to induce beneficial effects on human health [12]."

In addition, we have now added in the introduction which specific active compounds were found in our pomegranate extract: " The pomegranate extract used here was particularly rich in ellagitannins, and more specifically in punicalagins, as well as in ellagic acid." The exact quantities were added in the material and methods section (see below).

In the Materials and Methods section, the methods are properly presented, but the material section is missing.

I recommend a section with the material used

The material reference has now been added directly in the text.

Details of the supplement used: raw material used, method of preparation of the nutritional supplement, storage conditions, instructions for use.

Details about the raw material used have been added: "Each capsule of pomegranate extract contained at least 65mg punicalagins (>10%), 260mg polyphenols (>40%) and more than 13mg ellagic acid (>2%). The dose of 650mg was chosen based on the efficacy of this dose to improve muscle strength recovery after eccentric exercise [22]."

Details about the method of preparation of the nutritional supplement: "The capsules containing maltodextrin or pomegranate extract were prepared by a pharmacist."

Details about the storage conditions: "All products were stored in a closed room at a controlled temperature between 18 and 25°C and protected from light."

Details about the instructions for use: "Both groups were instructed (1) to consume 20g protein mixed into vegetable milk, orange juice or soup and (2) to ingest one capsule with a glass of water every day before lunch for 3 weeks."

The results are presented succinctly, but sufficiently.

In table 2, the statistical analysis is partially carried out. The standard deviation is not sufficient for statistical analysis.

Statistical analyses were performed by a professional statistician (acknowledged in the appropriate section), who used a linear mixed model for repeated measurements with subjects as random variable, and groups, visits, and their interaction as fixed independent variables. The model has been validated and used in many previous studies from the lab (just to cite a few: Warnier et al, Med Sci Sports Exerc, 2022; Warnier et al, Am J Physiol, 2022; van Doorslaer et al, FASEB J, 2021; Pachikian et al, Nutrients, 2021, Warnier et al, Sports, 2020; Pachikian et al, Blood Transf, 2020).

The discussion section needs to be significantly improved.

In this section, the obtained results must be commented and discussed

Lines 201-211 are not relevant to this section

We agree this kind of paragraph is debatable. Some referees specifically ask to recontextualize the discussion of the results, while others rather prefer to directly go into the discussion. We personally believe this kind of paragraph is valuable to better grasp the research question and the importance of the latter.

Line 229-232 - "Plasma potassium and urea levels increased while uric acid and total cholesterol decreased similarly in both groups, which is consistent with previous reports following protein intake in healthy and pathological populations" - please explain these results and their consequences. Please explain the results and their consequences for the following results as well.

The aim was to investigate safety and tolerance, but the expected benefits must be clearly established. We will not administer supplements just because they do not harm us. The goal is to do us good.

While we totally agree that we do not take supplements just because they do not harm us, but because they improve our health, the discussion about the potential health benefits will be possible only when a proper control group (no protein and no pomegranate) will be included. The design of the present study does not allow to clear conclusions about the beneficial health effects of our supplement. The only conclusion that can be drawn is about compliance and safety, not more. We will not hesitate to discuss the potential health benefits in detail in our following-up study that will aim at investigating those potential benefits.

We expect the reviewer may understand that each study design is drawn to answer one specific question, i.e. safety and tolerance of a supplement in a specific population. This preliminary study was required for ethical reasons before testing the potential beneficial health effects on a larger sample of older participants as the safety of the combination of protein and a pomegranate extract had never been tested before in people aged 65 years and more.

Round 2

Reviewer 2 Report

The authors responded to all my comments